# Outcomes of Mitral Valve Regurgitation Management after Expert Multidisciplinary Valve Team Evaluation

**DOI:** 10.3390/jcm13154487

**Published:** 2024-07-31

**Authors:** Myrthe J. M. Welman, Sebastian A. F. Streukens, Anass Mephtah, Loes P. Hoebers, Jindrich Vainer, Ralph Theunissen, Samuel Heuts, Jos G. Maessen, Patrique Segers, Kevin Vernooy, Arnoud W. J. van ‘t Hof, Peyman Sardari Nia, Pieter A. Vriesendorp

**Affiliations:** 1Department of Cardiology, Maastricht University Medical Centre+, P. Debyelaan 25, 6229 HX Maastricht, The Netherlands; bas.streukens@mumc.nl (S.A.F.S.); l.hoebers@zuyderland.nl (L.P.H.); j.vainer@mumc.nl (J.V.); ralph.theunissen@mumc.nl (R.T.); kevin.vernooy@mumc.nl (K.V.); arnoud.vant.hof@mumc.nl (A.W.J.v.‘t.H.); pieter.vriesendorp@mumc.nl (P.A.V.); 2Cardiovascular Research Institute Maastricht (CARIM), Maastricht University, Universiteitssingel 50, 6229 ER Maastricht, The Netherlands; sam.heuts@mumc.nl (S.H.); j.g.maessen@mumc.nl (J.G.M.); peyman.sardarinia@mumc.nl (P.S.N.); 3Faculty of Health, Medicine and Life Science, Maastricht University, Universiteitssingel 60, 6229 ER Maastricht, The Netherlands; a.meptah@student.maastrichtuniversity.nl; 4Department of Cardiology, Zuyderland Medical Centre, Henri Dunantstraat 5, 6419 PC Heerlen, The Netherlands; 5Department of Cardiothoracic Surgery, Maastricht University Medical Centre+, P. Debyelaan 25, 6229 HX Maastricht, The Netherlands; patrique.segers@mumc.nl

**Keywords:** multidisciplinary mitral valve team, multidisciplinary decision-making, mitral valve disease, mitral valve regurgitation, mitral valve management, Kaplan–Meier, conservative management mitral valve regurgitation

## Abstract

**Background/Objectives**: Mitral regurgitation (MR) affects millions worldwide, necessitating timely intervention. There are significant clinical challenges in the conservative management of MR, leaving a knowledge gap regarding the impact of multidisciplinary decision-making on treatment outcomes. This study aimed to provide insights into the impact of multidisciplinary decision-making on the survival outcomes of MR patients, focusing on conservative approaches. **Methods**: This study retrospectively analyzes 1365 patients evaluated by an expert multidisciplinary heart team (MDT) in a single center from 2015 to 2022. Treatments included surgery, catheter-based interventions, and conservative management. Propensity matching was utilized to compare surgery and conservative approaches. **Results**: Surgical intervention was associated with superior long-term survival outcomes compared to conservative and catheter-based treatments, particularly for degenerative MR (DMR). Survival rates of patients deemed by the MDT to have non-severe DMR were comparable to surgical patients (HR 1.07, 95% CI: 0.37–3.12, *p* = 0.90). However, non-severe functional MR (FMR) patients trended towards elevated mortality risk (HR 1.77, 95% CI: 0.94–3.31, *p* = 0.07). Pharmacological treatment for DMR was associated with significantly higher mortality compared to surgery (HR 8.0, 95% CI: 1.78–36.03, *p* = 0.001). Functional MR patients treated pharmacologically exhibited a non-significantly higher mortality risk compared to surgical intervention (HR 1.93, 95% CI: 0.77–4.77, *p* = 0.20). **Conclusions**: Survival analysis revealed significant benefits for surgical intervention, contrasting with elevated mortality risks associated with conservative management. “Watchful waiting” may be appropriate for non-severe DMR, while FMR may require closer monitoring. Further research is needed to assess the impact of regular follow-up or delayed surgery on survival rates, as pharmacological therapy has limited long-term efficacy for DMR.

## 1. Introduction

Mitral regurgitation (MR) stands out as a common valvular heart disorder, affecting 24.2 million people globally [1]. The prevalence of this valvular condition increases with age, is comparable across genders, and is associated with a worse prognosis [2]. MR can be classified by disease of the leaflets (degenerative or acute/subacute endocarditis) or diseases of the left atrium or left ventricle (functional), with a potential overlap between the two conditions. Untreated MR may lead to further adverse remodeling and progression to a severe disease state [3]. Since degenerative and functional MR are distinct pathologies, each requires a tailored approach and treatment strategy [4].

Surgical intervention for degenerative mitral regurgitation (DMR) is the cornerstone of management. The 2021 ESC/EACTS recommendations to treat heart valve disease advocate for surgery for symptomatic patients with severe DMR who are deemed to have an acceptable surgical risk by the heart team [5]. In addition, specific echocardiographic indicators and the presence of atrial fibrillation are considered indicators for intervention, irrespective of symptomatic status [4,5]. When these requirements are not satisfied, delaying surgery is considered a safe solution for asymptomatic patients with severe DMR, underscoring the importance of timely recognition and intervention [5,6].

Although surgical treatments for DMR have been studied for decades, extensive discussions remain regarding the optimal approach for individual patients. In relatively younger patients with DMR, surgical mitral valve repair often leads to high recovery rates and long-term survival [7,8]. To treat high-risk patients with DMR, several randomized trials are underway to study the effect of transcatheter edge-to-edge repair (TEER) compared to surgery [9]. Despite the well-known benefits of mitral valve surgery over pharmacological treatment, approximately 50% of eligible patients do not undergo the procedure [10]. However, a conservative approach alone in severe MR carries a poor prognosis, with a 1-year mortality rate reaching up to 20% and a 5-year mortality rate as high as 50% [11]. Meanwhile, there is limited evidence-based medical therapy for DMR to delay the progression of symptoms [5,6,12]. While data specific to patients with DMR with LV dysfunction are sparse, β-blockers and angiotensin-converting enzyme inhibitors can relieve symptoms once heart failure has developed and may help postpone the need for intervention [5,6,12,13].

The guidelines for functional mitral regurgitation (FMR), often caused by left ventricular remodeling, emphasize that guideline-directed medical therapy (GDMT) is the first step. If symptoms continue despite standard heart failure treatment, surgery should be considered immediately to avoid further loss of left ventricular systolic function or cardiac remodeling [5,6,13]. However, the optimal treatment for FMR is still debated due to its multifaceted nature, where FMR is one component of a complex disease process. Limited evidence suggests that surgical interventions following optimization of GDMT significantly improve survival outcomes for patients with FMR, making interventional management challenging [5,6,13,14,15,16]. Moreover, for patients with FMR, particularly those with ischemic heart disease, ongoing debate surrounds whether surgical mitral valve repair (MVP) or replacement (MVR) is the preferred treatment [8]. Additionally, TEER has emerged as a safe alternative for treating severe mitral regurgitation in patients with contraindications or high-risk surgical candidates [5,17].

Recognizing the complexity of mitral valve disease cases, multidisciplinary heart teams (MDTs) play a pivotal role in decision-making. Prior studies have demonstrated that decisions made by an expert MDT are associated with significantly improved survival outcomes, regardless of baseline patient characteristics, treatment modality, or the specific pathology of mitral valve disease [18]. Nevertheless, there are clinically still substantial challenges in the conservative management of MR [5,6,12,13,14,15,16]. Therefore, the objective is to give insights into the clinical decision-making of the MDT, specifically focusing on the conservative management of patients with DMR and FMR. This study aims to gain insight into how these decisions impact patient outcomes, particularly survival.

## 2. Materials and Methods

In this retrospective analysis, all patients whose mitral valve pathology was evaluated by the specialized mitral valve team at Maastricht University Medical Center (MUMC+) from 2014 to 2022 were included. These patients were referred from four regional hospitals or our center.

### 2.1. The Mitral Valve Heart Team Meetings

Our center has a specialized MDT that focuses on a balanced treatment strategy for the individual patient based on the specific mitral valve pathology, anatomic eligibility, comorbidities, background, and particular wishes [19]. The team is comprised of two cardiac surgeons, interventional cardiologists experienced in catheter-based mitral valve interventions, and an EACVI-certified imaging cardiologist. This team convened weekly, ensuring all members were present for the meetings only held when the team was complete. All referred patients initially underwent transthoracic echocardiography at the referring site. However, these echocardiograms were re-evaluated by our MDT as well to assess the severity and mechanism of MR. For patients assigned to surgical intervention, the feasibility of valve repair was critically judged. Isolated mitral valve repairs or replacements were assessed for suitability for an endoscopic approach based on anatomical eligibility. The MDT made treatment decisions for individual patients according to the latest guidelines [5,6].

### 2.2. Imaging Techniques

Patients underwent transthoracic echocardiography (TTE) at the referring hospital before evaluation. Those eligible for surgical or transcatheter mitral valve repair underwent three-dimensional (3D) transesophageal echocardiography (TEE), performed by our center’s imaging cardiologist. Additionally, patients with isolated mitral valve pathology deemed suitable for surgical intervention underwent computed tomography (CT) for 3D anatomical reconstruction of the aorta and peripheral vessels to assess eligibility for an endoscopic approach [20]. Coronary angiography (CAG) was conducted to evaluate potential concurrent coronary artery disease (either by CT or invasive angiography). Following this diagnostic imaging, patients were reassessed as needed to determine the treatment.

### 2.3. Mitral Valve Treatments

Beyond diagnosis and treatment planning, the expert MDT was integral to the treatment phase. Specialized imaging cardiologists provided intraoperative and post-procedural assessments, and the team ensured accurate postoperative evaluation and management. In instances of late complications or recurrent MR, patients underwent re-evaluation by the mitral valve heart team, with consideration given to potential additional therapies. The procedures were categorized into three groups: surgical interventions, catheter-based interventions, or conservative treatment. Surgical repair/replacement was performed at our center either via sternotomy or entirely endoscopically. An interventional cardiologist conducted percutaneous treatments including TEER, percutaneous indirect annuloplasty, or transapical chordal repair [21,22,23].

### 2.4. Endpoints

The primary endpoints were overall mortality and survival rates, affected by MDT decision-making. Evaluating overall mortality provided insights into the effectiveness of interventions in mitigating MR-related mortality risks. Survival rates offered a broader understanding of long-term outcomes following diverse clinical approaches recommended by the MDT.

### 2.5. Outcomes

Patient data were accurately collected from medical records. Clinical symptom severity and risk assessment were evaluated using the European System for Cardiac Operative Risk Evaluation (EuroSCORE) and the New York Heart Association dyspnea classification (NYHA), respectively. Echocardiographic features and the degree of MR were quantitatively assessed according to contemporary societal guidelines [24]. The primary endpoint, overall survival, was defined from the final date of the MDT meeting.

### 2.6. Statistical Analysis

Categorical variables were presented as absolute numbers with percentages. The normality of continuous variables was assessed through the Shapiro–Wilk test. Due to skewed distributions, all continuous variables were presented as median values with the first and third quartiles in parentheses. The chi-square test was performed to examine differences in patient baseline characteristics. The Kaplan–Meier method was utilized to plot survival probabilities over the 4-year follow-up period. The different procedures were distinguished for etiology and compared to the general population in a Kaplan–Meier analysis using survival data from the Human Life-Table Database [25]. Survival probabilities were calculated in percentages. Propensity score matching was conducted to compare the various conservative changes to surgery using covariates identified in the binary logistic regression model, and unknown data were not allowed. Propensity scores were matched using nearest-neighbor matching in a 1:1 ratio, with a caliper distance set at 0.20 [26]. Survival probability was analyzed after matching for baseline characteristics, EuroSCORE II, NYHA, left ventricular ejection fraction (LVEF), and valve etiology. The survival functions of the analyzed groups were compared using the log-rank test. Cox proportional hazards regression was employed to quantify the relative difference in survival over time using hazard ratios (HRs) with 95% confidence intervals (CIs). *p*-values < 0.05 were considered statistically significant. Statistical analysis was performed using R software v. 4.3.2. (R Foundation, Vienna, Austria).

## 3. Results

The retrospective analysis included 1365 patients who suffered from MR evaluated by the specialized mitral valve heart team at MUMC+. Within this cohort, 526 patients underwent surgical intervention, 186 underwent catheter-based procedures, and conservative management was chosen for 653 patients. For patients deemed too fragile by the MDT, an outpatient clinic visit was performed in 36% of cases to validate the decision. Within the pharmacological approach for 123 patients, the MDT considered GDMT alone to be sufficient. The MDT determined that for 328 patients, the MR was not severe enough to necessitate intervention, while 51 patients autonomously opted for conservative management. The final decision pathway of the patients is demonstrated in Figure 1.

Patients selected for surgery were younger, had lower EuroSCORE-based risk, and had fewer comorbidities. Patients who underwent surgical intervention had higher LVEF at baseline. Degenerative MR was prevalent among most patients in the surgery and catheter-based intervention groups, whereas FMR predominated in the conservative treatment group. Detailed baseline patient characteristics and echocardiographic parameters are presented in Table 1.

The Kaplan–Meier test was utilized to estimate survival, with a median follow-up of 950 (337–1673) days. In general, all patients discussed in the expert MDT exhibited higher mortality rates compared to the overall population, irrespective of the etiology of the mitral valve. The general population aged 72 years had a 4-year survival probability of 89.39%. Patients with DMR had 4-year survival probabilities of 83.68% for surgery, 67.68% for catheter-based interventions, and 57.16% for conservative management (Figure 2a). Survival probabilities over 4 years for patients with FMR were 80.81% for surgery, 50.97% for catheter-based interventions, and 60.00% for conservative management (Figure 2b).

The estimated survival rate was assessed to determine the prognosis among patients in the conservative group. Patients deemed to be at high risk or contraindicated for all other treatments had a 4-year survival probability of 36.67%. Those with non-severe MR exhibited a 4-year survival probability of 71.41%, while patients for whom GDMT was considered sufficient demonstrated a survival probability of 55.72%. Patients who opted for pharmacological therapy themselves demonstrated a 4-year survival probability of 41.82% (Figure 3a). After comparative analysis with propensity matching between surgical and conservative approaches, except LVEF for surgery versus non-severe MR, there was no significant difference in baseline patient characteristics or echo parameters (Table A1 and Table A2). Survival analysis revealed that patients who autonomously chose their treatment path faced a higher mortality risk than their surgically managed counterparts (HR 4.47, 95% CI: 1.29–15.46, *p* < 0.01, Figure 3b).

Additionally, propensity analysis indicated that the survival rates of patients with non-severe DMR were comparable to those of patients undergoing surgery (HR 1.07, 95% CI: 0.37–3.12, *p* = 0.90, Figure 4a). Non-severe FMR patients exhibited a trend of elevated mortality risk compared to their surgical counterparts, though this was not statistically significant (HR 1.77, 95% CI: 0.94–3.31, *p* = 0.07, Figure 4b). Moreover, individuals under pharmacological treatment for DMR exhibited a significantly higher mortality risk compared to those who underwent surgery (HR 8.0, 95% CI: 1.78–36.03, *p* = 0.001, Figure 4c). Reasons by the MDT for pharmacologically managing patients with DMR included unsuitability for surgery due to anatomical considerations, postponement of surgery due to other medical priorities, stable DMR, or few symptoms in combination with advanced age. Individuals under medication treatment for FMR exhibited a non-significant higher mortality risk compared to those who underwent surgery (HR 1.93, 95% CI: 0.77–4.77, *p* = 0.20, Figure 4d).

## 4. Discussion

Severe MR is a complex condition requiring a multidisciplinary approach for optimal treatment, preferably with a specialized MDT [18]. Despite comprehensive guidelines on the role of conservative treatment, there are still challenges in the clinic about the most effective approach to managing MR [5,6,12,13,14,15,16]. Therefore, this study aimed to give insights into the clinical decision-making processes of multidisciplinary teams, specifically focusing on the conservative management of patients with the distinction of both MR etiologies, regarding survival. Key findings of this study reveal that surgical intervention was associated with the best outcome in patients with severe MR, yielding excellent long-term outcomes, especially in DMR. While patients with non-severe DMR with conservative treatment demonstrate similar survival prospects as those opting for surgery, individuals with non-severe FMR tend towards elevated mortality risk compared to surgical intervention. Patients receiving pharmacological therapy for DMR presented with a significantly higher mortality risk compared to their surgically treated counterparts.

### 4.1. General Overview of Treatments for Degenerative and Functional Mitral Valve Regurgitation

At the forefront, all patients discussed within the expert MDT exhibit elevated mortality rates compared to the general population, with FMR revealing the least favorable survival prospects. This features the increased mortality risk inherent to severe FMR, emphasizing the imperative vigilant surveillance, efficacious therapeutic interventions, and potential preventive measures to improve health outcomes for these individuals. In this cohort, DMR demonstrated a higher average survival rate across all treatments compared to FMR, consistent with previous studies indicating a poorer prognosis [13,27]. This is attributable to the fact that FMR is predominantly a consequence of ventricular remodeling rather than being primarily attributed to the valve itself. Our results indicated that both DMR and FMR cases revealed the highest long-term survival rates with a surgical approach. These results support the 2021 ESC/EACTS guidelines for the treatment of heart valve disease suggesting that timely surgical intervention is deemed essential for patients suffering from severe MR, regardless of their etiology [5]. The findings indicate that severe FMR may indeed yield higher survival rates when comprehensively discussed within the MDT, with consideration given to potential surgical interventions following clinical guidelines. Within DMR, patients who underwent catheter-based interventions had the second-highest survival rates, suggesting that catheter-based interventions are a viable alternative for severe DMR, tailoring patients confronted with surgical contraindications or elevated operative risks. However, in the case of FMR, the catheter-based interventions group drops below the conservative group after approximately two years. Patients with FMR often experience heart failure, and those undergoing catheter-based interventions are typically high-risk, which can impact long-term survival. In this study, the 60% survival rate for patients with FMR who were treated pharmacologically, highlights the severity of the condition, and the complexity of optimized medical therapies [13]. It is plausible that the conservative group with FMR fares better in terms of survival compared to the patients who underwent catheter-based interventions due to the substantial diversity among conservative individuals. In addition, our data revealed that DMR patients who received pharmacological treatment exhibited the highest mortality rates within this etiology. Optimal GDMT may alleviate symptoms and prevent heart failure but does not provide a long-term solution for mitral valve disease [10].

### 4.2. Conservative Treatment in General

Guidelines suggest that GDMT should be considered in DMR and FMR [5,6]. The conservative group in this cohort study represents a diverse group. The results demonstrated that the expert MDT predominantly opted for conservative management of FMR. Additionally, patients treated conservatively exhibited significantly lower LVEF than those undergoing surgery or catheter-based procedures. Moreover, most patients were diagnosed with grade II MR compared to those undergoing catheter-based procedures or surgery. Considering our findings, patients must be informed about the potential consequences of deviating from the advice of the MDT based on the guidelines for severe MR. Specifically, our results indicate that patients who autonomously opted for conservative treatment face a significantly higher mortality risk. This suggests the importance of transparent and comprehensive communication between healthcare providers and patients regarding treatment options, risks, and potential outcomes.

### 4.3. Conservative Management in Degenerative and Functional Mitral Regurgitation

This cohort study revealed comparable survival rates between the patients for whom the MDT decided that the DMR was not yet severe enough and those who underwent surgery for this condition. These patients were referred to the MDT for discussion by specialists due to a suspicion of severe MR. For asymptomatic patients with severe DMR and no left ventricular dysfunction, two guidelines establish different criteria for surgery. While the ACC/AHA guideline recommends MVR based on success rate and mortality criteria, the ESC/EACTS guideline validates a strategy of “watchful waiting” unless there is the presence of atrial fibrillation or elevated systolic pulmonary arterial pressure [16]. These results support the ESC/EACTS guideline recommending “watchful waiting” as a safe strategy for asymptomatic patients with severe DMR, ideally managed in a specialized MDT [5]. Additionally, the results indicated that pharmacological treatment for DMR is associated with significantly extremely higher mortality compared to surgical intervention. This suggests that pharmacological treatment may not be a solution for DMR in terms of survival. An intervention might be a more effective option in certain cases of severe DMR and could be worth considering, despite the associated risks [28]. However, it is important to note that the decision of the expert MDT between pharmacological treatment and surgical intervention depends on several factors, including the overall health status of the patient and individual treatment preferences.

This cohort analysis found a trend between patients whose MDT determined that FMR was non-severe and those who underwent surgery for the disease. Over approximately one year, a discernible dichotomy emerged between the two treatments. In addition, after contrasting the survival rates of patients managed pharmacologically and those subjected to surgical intervention for FMR, our observations revealed the poorest survival rates for the pharmacological treatment compared to surgery in FMR. Notably, a disadvantage divergence emerged again after a year. Although there is a difference in survival rates between patients undergoing surgery and those receiving pharmacological treatment, this difference is non-significant, suggesting that the decision of the expert MDT to reject surgery for the patient may have been justified. This emphasizes the need for the MDT to carefully consider all available alternatives while considering the patient’s unique circumstances and demands. Given that our results revealed distinctions between FMR and surgery in both circumstances, perhaps not only “watchful waiting” should be considered in FMR, but also proactive re-evaluation by the MDT.

### 4.4. Limitations

In this single-center study, we aimed to evaluate the impact of the decisions of the expert MDT regarding survival. Ethical constraints prevent conducting a randomized trial on the heart team. Thus, our research group has focused on a historical cohort to explore potential evidence for the advantage of this dedicated MDT method. Propensity score matching was employed to facilitate an accurate comparison of survival outcomes in conservative management compared to those who underwent surgery. However, there is still a possibility of unmeasured confounders that could influence the outcomes. The causes of death are speculative, and there is no evidence that improved follow-up, including the possibility of later surgery, provided higher survival rates. Furthermore, given the period of several years over which the study was performed, advancements in technology and changes in clinical practices may also impact the results. Nevertheless, a notable strength of this study is its comprehensive survival analysis over four years, while many other studies only offer short-term follow-up.

## 5. Conclusions

This study provides a comprehensive overview of decisions made by an expert MDT, revealing that surgical intervention yields superior long-term survival outcomes compared to conservative and catheter-based treatments, especially for DMR, resulting in superior long-term outcomes. Meanwhile, there could be potential benefits from a surgical approach for FMR patients, deliberated within the MDT and aligned with clinical guidelines. Conservative management carries an increased risk of mortality compared to surgical interventions. The management of MR should be tailored to the specific etiology of the condition. While “watchful waiting” may be suitable for non-severe DMR, FMR may require more attention through regular follow-up and re-evaluation. However, further research is warranted to underestimate whether regular follow-up or delayed surgery might result in improved survival rates. Additionally, pharmacological therapy has limited efficacy as a long-term solution for DMR.

## Figures and Tables

**Figure 1 jcm-13-04487-f001:**
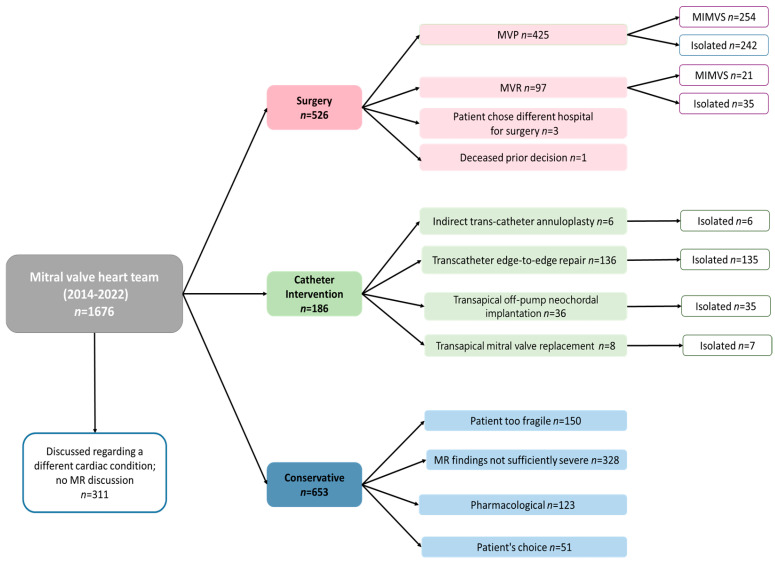
Flowchart decisions in mitral valve heart team.

**Figure 2 jcm-13-04487-f002:**
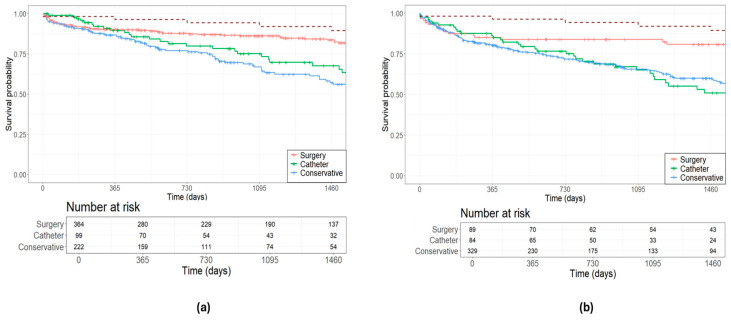
Estimated survival outcomes of treatment decisions in (**a**) degenerative mitral valve disease; and (**b**) functional mitral valve disease. The red dotted line represents the general population.

**Figure 3 jcm-13-04487-f003:**
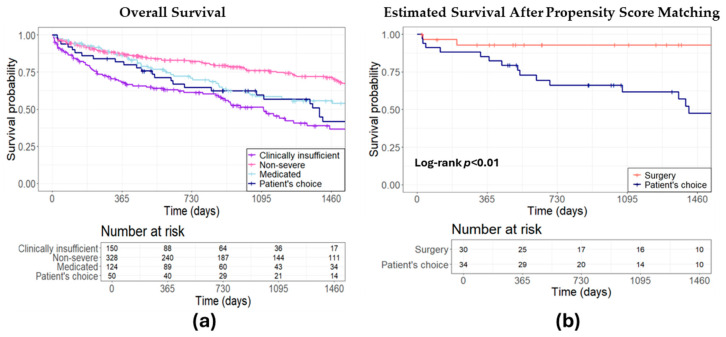
Estimated survival: a comparative analysis of conservative treatment strategies. (**a**) Overall survival comparison for different conservative approaches; (**b**) surgical intervention versus patient’s autonomous choice for conservative treatment.

**Figure 4 jcm-13-04487-f004:**
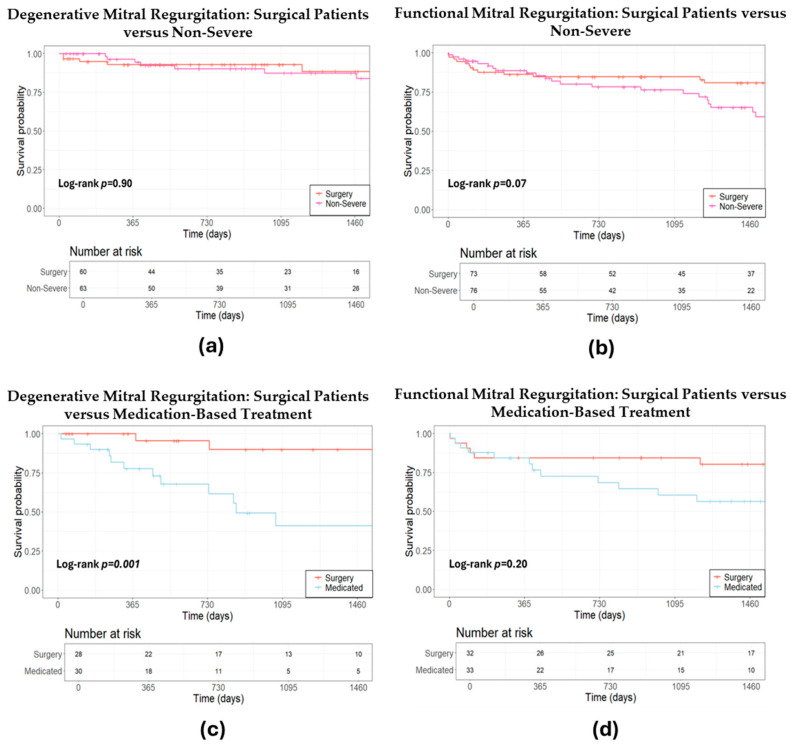
Estimated survival after propensity score matching: a comparative analysis of conservative treatment strategies versus surgery. (**a**) Estimated survival after propensity score matching: surgical versus non-severe degenerative mitral regurgitation; (**b**) estimated survival after propensity score matching: surgical versus non-severe functional mitral regurgitation; (**c**) estimated survival after propensity score matching: surgical versus medication-based treatment for degenerative mitral regurgitation; (**d**) estimated survival after propensity score matching: surgical versus medication-based treatment for functional mitral regurgitation.

**Table 1 jcm-13-04487-t001:** Clinical and echocardiographic characteristics at the time of multidisciplinary heart teams. Data are characterized as *n* (%) and median (IQR).

	Surgery *n* = 526	Catheter *n* = 186	Conservative *n* = 653	Total *p*-Value
Age (years)	67.77 (60.4–3.0)	78.48 (71.7–82.3)	74.75 (67.2–80.7)	<0.001
Gender (male)	329 (55.90)	104 (55.91)	338 (51.84)	<0.001
BMI (kg/m^2^)	25.35 (23.1–28.0)	24.82 (22.29–28.25)	25.79 (22.7–28.2)	0.396
Diabetes	45 (8.56)	29 (15.59)	123 (18.84)	<0.001
EuroSCORE II	1.97 (1.0–4.0)	2.8 (1.81–5.1)	2.32 (1.3–4.3)	<0.001
NYHA classification				0.002
I	142 (27.0)	26 (14.0)	178 (27.3)	
II	216 (41.1)	82 (44.1)	282 (43.3)	
III	147 (28.0)	70 (37.6)	155 (23.8)	
IV	12 (2.3)	1 (0.5)	15 (2.3)	
Unknown	9 (1.6)	7 (3.8)	23 (3.3)	
PHT				<0.001
Moderate	194 (36.9)	49 (26.3)	123 (18.9)	
Severe	40 (7.6)	16 (8.6)	50 (7.6)	
Atrial fibrillation	201 (38.3)	82 (44.1)	276 (42.3)	0.401
Kidney dysfunction				<0.001
Normal	223 (42.6)	125 (67.2)	467 (72.3)	
Moderate	255 (48.6)	41 (22.0)	123 (19.1)	
Severe	45 (8.6)	17 (9.1)	46 (7.2)	
Dialysis	1 (0.2)	1 (0.5)	5 (0.8)	
Unknown	0	2 (1.2)	6 (0.9)	
LVEF (%)				<0.001
Good (>50)	387 (73.6)	92 (49.5)	321 (49.2)	
Moderate (31–50)	125 (23.7)	64 (34.4)	216 (33.1)	
Poor (21–30)	11 (2.1)	21 (11.3)	84 (12.9)	
Extremely poor (<21)	3 (0.6)	9 (4.8)	31 (4.8)	
MR severity				<0.001
Grade I	0		47 (7.2)	
Grade II	57 (10.9)	14 (7.5)	295 (45.2)	
Grade III	86 (16.3)	45 (24.2)	134 (20.6)	
Grade IV	383 (72.8)	127 (68.3)	176 (27)	
Classification valve disease				<0.001
Degenerative	367 (69.8)	99 (53.2)	222 (34.0)	
Functional	92 (17.5)	84 (45.2)	330 (50.6)	
Rheumatic	26 (4.9)	0	30 (4.6)	
Endocarditis	8 (1.5)	0	9 (1.4)	
Combined	6 (1.1)	2 (1.1)	6 (1.0)	
Stenosis	13 (2.5)	0	35 (5.4)	
Other	14 (2.7)	1 (0.5)	20 (3.0)	
LVESD (mm)	4.0 (2.0–8.0)	6.0 (4.0–11.0)	4.0 (2.0–8.0)	<0.001
LAVi	6.0 (3.0–12.0)	9.0 (6.0–15.0)	6.00 (3.0–15.0)	<0.001

BMI: body mass index, EuroSCORE: European system for cardiac operative risk evaluation, LAVi: left atrial volume index, LVESD: left ventricular end-systolic diameter, LVEF: left ventricular ejection fraction, MR: mitral regurgitation, NYHA: New York Heart Association classification for dyspnea, PHT: pulmonary hypertension.

## Data Availability

The corresponding author will disclose the data used in this article upon reasonable request.

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
