# Peer review of "Outcomes of Mitral Valve Regurgitation Management after Expert Multidisciplinary Valve Team Evaluation"

_jcm, 2024, doi:10.3390/jcm13154487_

Round 1

Reviewer 1 Report

Comments and Suggestions for Authors

I read with great interest, this paper on a formal MDT for mitral valve workup

The paper is well written and well organized. I think it substancially contributes to our field, as we need to standardize MDT across all institutions. 

Clearly, there are limitations of the analysis side (i.e.- surgery patients healthier)... however, the limitations are described well by the authors.

Excellent paper.

Author Response

We would thank the reviewer for the kind comments.

Reviewer 2 Report

Comments and Suggestions for Authors

In their article "Outcomes of Mitral Valve Regurgitation Management after Expert Multidisciplinary Valve Team Evaluation", Welman et al. evaluated the impact of decision-making by a multidisciplinary team (MDT) on the survival of patients with mitral regurgitation (MR), comparing conservative treatment, surgery and catheter-based interventions. The study retrospectively analyzed data from 1365 patients treated between 2015 and 2022. The study concludes that surgery offers a better long-term survival rate, especially in degenerative MR, while conservative treatment is associated with a higher mortality rate.

In general, it is a well-written, well-structured and well-presented study. I just have a few minor questions.

1. what happens to the patients who were selected as the "conservative" group because MR was not sufficient?

2. how can LVEF be unknown in 9 patients (Table 1) if TTE and/or TEE is required to determine MR grade?

3. how many patients were considered for surgery because they had conditions that contraindicate an interventricular approach, such as active endocardia? Was there a difference in risk and outcome in this patient group?

The authors describe a important approach to decision making in the treatment of mitral valve insufficiency.

Thank you for the opportunity to review this article.

Author Response

  1. what happens to the patients who were selected as the "conservative" group because MR was not sufficient?
     During the follow-up phase, 0.4% of patients were deemed too frail, 0.3% of patients whose mitral regurgitation was not severe enough to warrant surgery, 0.7% of patients treated pharmacologically, and 0.2% of patients who opted for conservative management were eventually operated on in the follow-up phase. This indicates that nearly all patients remained under conservative treatment. While no significant difference in survival was observed, the distribution of patients in this comparison is too disproportionate. Consequently, we decided not to include these results in the study.

  1. how can LVEF be unknown in 9 patients (Table 1) if TTE and/or TEE is required to determine MR grade?
    Thank you for your thoughtful comment. We have reiewed the echocardiograms and had the pre-intervention echo reassessed by a specialist. This has been corrected in Table 1.

  1. how many patients were considered for surgery because they had conditions that contraindicate an interventricular approach, such as active endocardia? Was there a difference in risk and outcome in this patient group?
    Thank you for your interesting question. In our raw data, we observed a trend (p=0.06) toward the management decision of surgery versus conservative treatment for endocarditis as decided by the MDT. However, due to the very small study population (n=9 without propensity matching), we recognize the need to include more patients to make a definitive conclusion. Unfortunately, expanding our sample size for this study was not feasible, but we will keep this in mind for future research. 

Reviewer 3 Report

Comments and Suggestions for Authors

I read with interest the article by Dr Welman et al. This was an attempt to evaluate outcomes of Mitral Valve Regurgitation Management after expert Multidisciplinary Valve Team Evaluation.

I have a few major comments.

What was striking throughout the manuscript is that the authors erroneously present the two mechanisms of MR, degenerative or primary and functional or secondary as a common pathway of assessment, approach and treatment. This does not hold true as each pathology has very discreet characteristics and different therapeutic approaches. Optimal medical treatment is certainly the cornerstone of first-line management in secondary MR and the patients that are persistently symptomatic they are further referred for advanced therapeutic options including surgery or TEER. On the other hand, cases with symptomatic severe primary MR are eminently treated with surgery or TEER depending on multiple factors including anatomical considerations. Asymptomatic cases of severe primary MR that meet certain criteria (LVEF, LVESD, AF, PASP...) are still directed for surgery with little if any room for medical management. This is not clearly documented anywhere in the text.

It also comes to the attention that on multiple occasions the authors report paucity of supporting data or clear guidelines. This certainly does not hold true as over the past at least decade lots of experience, data and knowledge have accrued and led to quite robust international guidelines.

Additionally, I am not entirely sure why in the manuscript authors report non-severe MR treatment. As per ESC guidelines, only cases with severe MR are potentially further considered for interventional options.

One other remark is that the authors do not clearly document the endpoints of their study and this is a serious omission from the Materials and Methods section.

I have many comments on the results but the most important one is that the authors conclude that medical management for primary MR leads to higher mortality compared to surgery. The guidelines clearly suggest surgical repair ideally or replacement if repair is not feasible in severe primary MR. This bears the question why there were patients in their cohort with severe primary MR that surgery was not offered to them and hence unsurprisingly had worse outcomes. Certainly, there are cases where no intervention modality is an option for example due to limited prognosis due to other conditions (cancer with poor prognosis, significant frailty...) and we encounter such cases in daily clinical practice. Overall though, it is not appropriate to send wrong messages to the readers.

Because of the above, unfortunately, I cannot see the novelty in this manuscript and although I have to commend the authors for keeping such a robust structure in their heart team meetings, the evidence is already out there and I doubt there is any relevant gap that this article would fill out.

Comments on the Quality of English Language

Minor editing needed, overall acceptable.

Round 2

Reviewer 3 Report

Comments and Suggestions for Authors

I thank the authors for the changes they made to the content. I believe this is a significant improvement to the originally submitted article.

I would support publication of the manuscript.

Comments on the Quality of English Language

Use of English language is appropriate